# Role of Sex and Age in Fatal Outcomes of COVID-19: Women and Older Centenarians Are More Resilient

**DOI:** 10.3390/ijms24032638

**Published:** 2023-01-30

**Authors:** Calogero Caruso, Gabriella Marcon, Giulia Accardi, Anna Aiello, Anna Calabrò, Mattia Emanuela Ligotti, Mauro Tettamanti, Claudio Franceschi, Giuseppina Candore

**Affiliations:** 1Laboratorio di Immunopatologia e Immunosenescenza, Dipartimento di Biomedicina, Neuroscienze e Diagnostica Avanzata, Università di Palermo, 90134 Palermo, Italy; 2Dipartimento di Scienze Medico Chirurgiche e della Salute, Università di Trieste, 34149 Trieste, Italy; 3Dipartinento di Area Medica, Università di Udine, 33100 Udine, Italy; 4Laboratorio di Epidemiologia Geriatrica, Istituto di Ricerche Farmacologiche Mario Negri IRCCS, 20156 Milano, Italy; 5Dipartimento di Scienze Mediche e Chirurgiche, Alma Mater Studiorum, Università di Bologna, 40126 Bologna, Italy

**Keywords:** age, COVID-19, immune responses, longevity, mortality, narrative review, sex, Spanish flu

## Abstract

In the present paper, we have analysed the role of age and sex in the fatal outcome of COVID-19, as there are conflicting results in the literature. As such, we have answered three controversial questions regarding this aspect of the COVID-19 pandemic: (1) Have women been more resilient than men? (2) Did centenarians die less than the remaining older people? (3) Were older centenarians more resistant to SARS-CoV-2 than younger centenarians? The literature review demonstrated that: (1) it is women who are more resilient, in agreement with data showing that women live longer than men even during severe famines and epidemics; however, there are conflicting data regarding centenarian men; (2) centenarians overall did not die less than remaining older people, likely linked to their frailty; (3) in the first pandemic wave of 2020, centenarians > 101 years old (i.e., born before 1919), but not “younger centenarians”, have been more resilient to COVID-19 and this may be related to the 1918 Spanish flu epidemic, although it is unclear what the mechanisms might be involved.

## 1. Introduction

SARS-CoV-2, a Betacoronavirus (β-CoV) of the Coronaviridae family, is transmitted via inhalation or direct contact with droplets from infected individuals. To infect lung alveolar epithelial cells, it uses, as the entry receptor, the human angiotensin II converting enzyme (ACE2) and, for priming, the transmembrane serine protease 2 (TMPRSS2), which cleaves the virus spike. This β-CoV infects regardless of ethnicity, age, and sex, but with several clinical outcomes. Clinical manifestations, indeed, can range from mild to severe respiratory disease. Most cases of COVID-19 have a mild symptomatic disease with a moderate mortality rate, but older patients have a higher risk of contracting a severe disease, with hospitalization and death [1]. However, several factors influence susceptibility to and the severity of COVID-19: (1) Lifestyle, referring to habits (i.e., food quality, physical activity, smoking, and drug use), as well as socio-economic parameters (i.e., level of education, ability to access high-quality medical information, and possibilities of access to the health system), that is, the social determinants of health. (2) Environment intended not only how as exposure to xenobiotics, but also as exposure to multiple pathogens. (3) Genetics, in particular the genetics of immune responses. (4) Viral genetic variants. (5) Age, sex/gender, and the presence of comorbidities [2].

In the present review, we will focus on the role of age and sex in the deadly outcome of COVID-19, since in the literature, as discussed below, there are conflicting results. We will focus on sex, and we will touch on the possible role of gender in the discussion, as well as the potential role of genetics and lifestyle in sex and gender different outcomes. So, we will try to answer three disputed questions concerning the COVID-19 pandemic: (1) Have women been more resilient than men? (2) Did centenarians die less than remaining older people? (3) In the first pandemic wave of 2020, were older centenarians (i.e., centenarians > 101 years old, born before 1919) less resistant to the SARS-CoV-2 compared to younger (100 and 101 years old, born after 1919) centenarians? The first question refers to findings demonstrating that women live longer than men even during severe famines and epidemics [3]. The second and third questions refer to the well-known datum that centenarians are successful agers compared to other older adults [4,5,6], and to findings showing a different resilience in “younger” and “older” centenarians when centenarian mortality was analysed by date of birth [7,8]. In the context of susceptibility to age, the possible role of comorbidities will also be mentioned.

To carry out this narrative review, we queried the PUBMED database. Regarding mortality due to sex differences, we included the terms Sex Differences, COVID-19, and Death (or Mortality), limiting ourselves to meta-analysis and reviews to avoid analysing a plethora of studies and obtained seven entries. Of these, only three were relevant to our research. Analysing, then, both papers cited by these three articles and the list of papers citing them, we found three more pertinent reviews (plus two other papers of some interest). Due to the overlap of the analysed data, only three reviews are discussed in the following paragraphs [9,10,11]. Moreover, since Italy was the first country affected by SARS-CoV-2 in the western world, we have added a paper on Italian data in the first months of the epidemic [12]. We have also added a paper on a study carried out in the Netherlands because it is very informative as it shows comparisons with other infectious diseases [13]. As regards the mortality of centenarians, we entered the terms Centenarian, COVID-19, and Death (or Mortality), obtaining eight entries. Of these, only four were pertinent to our research. Analysing, then, both the papers cited by these four articles and the list of papers citing them, we found a fifth paper pertinent. All five articles [7,8,14,15,16] are discussed in the following paragraphs.

## 2. Have Women Been More Resilient Than Men?

A study on sex differences in COVID-19 mortality rates in the first months of the pandemic in Italy demonstrated that mortality rates were higher for men of all ages and reached values that were more than three times those of women aged 50–69 [12]. Data from different countries such as Peru, Spain, England, France, the USA, Mexico, and Sweden, updated in July 2020, confirmed that male mortality was higher than females [10]. Then, Scully et al. [9] reviewed data on COVID-19 case fatality rates (CFRs) across 38 countries. The COVID-19 CFR increased both for men and women with advancing age, but men had a significantly higher CFR than women at all ages from 30 years with an average male CFR 1.7 times significantly greater than the average female. The data had been updated in May 2020.

A meta-analysis based on 34 studies published in 2020 demonstrated that the mortality rate of hospitalized patients was 6.6%, with men having significantly higher mortality than women, whereas there was no significant difference in the ratio of admission to intensive care units [11].

In a further study performed in the Netherlands [13] authors compared the male–female population mortality rate ratio of COVID-19 cases to the overall population mortality rate ratio, the mortality rate related to infectious diseases in general as well as pneumonia, and influenza mortality rate. The authors calculated general population-specific mortality over the previous five years (2015–2019). Although the ratio increased in all examined infectious diseases, was on average higher in COVID-19 patients, hence the authors suggested that additional COVID-19-specific mechanisms should contribute to this mortality discrepancy. The ratio remained significant after adjustment for age and comorbidities. In conclusion, the paper confirmed male sex as a predisposing factor for severe outcomes of COVID-19, independent of comorbidities.

So, it appears that women were more resilient than men. However, Marcon’s data discussed in the next paragraph do not agree with this statement, suggesting that male centenarians are more resilient than female centenarians [14].

## 3. Did Centenarians Die Less Than Remaining Older People?

As previously stated, Italy was the first European country to be affected by COVID-19. The biggest cluster of cases occurred in Lombardy, the most populous Italian region, and older people were hit in the hardest way. In this population, Marcon et al. [14] wondered if (a) the COVID-19 mortality in centenarians was lower than that in less-aged people, and (b) the sex differences in mortality in different age classes. Comparisons were made using total mortality (i.e., not only confirmed COVID-19 cases) at the peak of infection (March 2020) against March’s total mortality of previous years. They did not find a decreased mortality in centenarians relative to younger ages but highlighted a difference between sexes across different age classes: while mortality in “younger” ages was much higher in men than in women, the rate at which the risk increased by age was slower in men than in women, so that centenarian women had a higher mortality rate. They suggested that the pro-inflammatory status of older people, inflamm-ageing, i.e., the conceptualization of the immune changes in response to lifelong stress [17], could explain such age-related vulnerability.

A reason for the higher female centenarian mortality than men centenarian mortality might have to do with the “female-male health-paradox”, i.e., centenarian men are fewer but more selected, healthier, and resilient than centenarian women [18,19]. Paradoxically, women live longer than men and appear stronger during famines and epidemics [3], but they show worse health than men when they become centenarians. This characteristic has been observed in different studies worldwide [18] and it is also evident in Northern Italy, including Lombardy, and in North-East of Italy, where the women in the “Centenarians in Trieste Study” suffer greater morbidity, but they live longer than men reaching 90% prevalence [14].

On the other hand, the data obtained from a low number of very well-characterized subjects of “Centenarians in Trieste Study” suggest that both centenarian women and men looked strong during the peak of the COVID-19 pandemic even though the pandemic profoundly challenged the entire health system and care of older people. That might depend on the different strategies used by the Italian Regional Health Services (RHS) in the single different territories since in Trieste the RHS is organized around micro-areas which can have closer contact with patients: frailer people (oldest-old and centenarians have a rising proportion of frailty [20]) can benefit most from this structure. Therefore, this higher presence near the patients themselves in Trieste than in Lombardy would have a greater effect on frailer people, hence in centenarians [14].

In Sicily (Italy), Caruso et al. [8] compared the 2020 mortality data of the >91 years old people, divided by sex and date of birth, to 2019 data as a control. The crude excess mortality between the two years was calculated as a percentage. Looking at the >91 years old people, the excess mortality of these subjects was higher than in the general population, being 11% (8140 vs. 7352), higher in men than in women (13% vs. 10%). As expected, this excess mortality was higher than those observed in the Sicilian population that was 6% (55,583 deaths compared to 52,405) Examining centenarians separately the excess mortality raised to 28% and, again, was higher in men than in women (33% vs. 27%). These data confirm that women are more resilient to the virus than men as suggested by anecdotal findings discussed below [21,22], but not by the study conducted in Lombardy [14].

The different data observed and reported at a higher magnification and higher granularity in single geographical areas and cities, institutions, and settings are also the consequence of the basic heterogeneity of the ageing phenotype, which is particularly evident at the extreme ages and suggests that outcomes may differ by robustness or other characteristics of the individual and are always highly diverse and context-dependent [23]. Another difference from Sicily is represented by the fact that Lombardy was the most affected region in Italy, and in that study, the deaths were analysed at the peak (month, not year) of the epidemic [14].

A retrospective multicenter cohort study was performed by Couderc et al. [15] using data from 15 nursing homes located in the Marseille area, enrolling older residents with confirmed COVID-19 between March and June 2020. The mortality rate was 50% among the 12 centenarians, higher than that observed among the other residents (median age 89 years; 24.6%).

In a study performed in German Long-Term Care Facilities, [16] lower rates of COVID-19–relevant hospital admissions in centenarians than in the oldest (>80 years) were observed. However, COVID-19 hospital mortality was significantly higher in female centenarians compared to female non-centenarians. Despite similar percentage values, it was not significant for centenarian males given the higher mortality among the oldest males.

So, it seems that centenarians did not die less than the remaining older people. However, it must point out that a study performed in Belgium showed that the overall death rate from COVID-19 was 130 times higher among nursing home residents than outside, due to the near multiplicative effects of differences in the resident age and sex structure, health frailty, and infection risk. So, the Authors suggested that in epidemiological studies, nursing home residents, including centenarians, should be treated as a very specific population due to their extreme vulnerability to COVID-19 [24].

There is a big exception to what we are writing about centenarians and mortality from COVID-19. The COVID-19 pandemic appears, indeed, to have increased longevity in Japanese centenarians. The projected number of centenarians increased from 37,005 in 2019 to 41,802 in the following year. It is noteworthy that the number of pneumonia deaths was 64,965 from January to September 2020, a decrease of 13,950 deaths from the same period a year earlier. The number of flu deaths was 941, a decrease of 2340 from the previous year. Such a decrease in deaths from infectious diseases that obviously affected all segments of the older population, including centenarians, probably depends on the characteristics of Japanese society [25]. There are many theories to explain the low number of COVID-19 cases in Japan, but we do not have enough information to confidently determine the cause of this surprising discrepancy. It is certainly true that Japanese customs do not involve shaking hands, hugging, or kissing when greeting. Additionally, many Japanese people wore cloth or paper masks even before the pandemic. Based on Japan’s traditional approach to social relationships and infectious disease control, there has been an early and unconditioned adoption of the “3Cs” (avoiding enclosed spaces, close contact, and close conversations). This approach anticipated what physicians and scientists later learned about the airborne nature of SARS-CoV-2 transmission [26,27].

## 4. Older Centenarians Were More Resistant to the SARS-CoV-2 Compared to Younger Centenarians?

Thus, previous studies show that centenarians were not particularly resilient to COVID-19. However, anecdotal data suggested the resilience to COVID-19 of either centenarian, regardless of sex, or only male centenarians [21,22,28]. Overestimating these anecdotal data is a cognitive bias: we are struck by the data of survival, but not of mortality. However, it is intriguing that most of the observations are for “older” centenarian women.

In fact, there are several older (>101 years old) centenarians, mostly women, who fell ill with COVID-19 and recovered spontaneously or after a short hospitalization [28,29]. Then, Maria Concetta La Mensa (born 14 December 1913), an Italian semi-supercentenarian, had a minimally symptomatic infection in November 2020 and was vaccinated in 2021 (Caruso, Ligotti and Candore, personal observation). Maria “Julia” Van Hool (7 May 1909–29 April 2021) was a Belgian supercentenarian who tested positive for COVID-19 and tested negative within a couple of weeks [30]. Iris Estay was born in Quilota, Chile on 3 September 1910, and tested positive for COVID-19, she became negative in a couple of weeks. After a few months, she had a new infection and recovered after a few days [31]. Recently, three cases of Brazilian supercentenarians (a 114-year-old woman and two 111- and 110-year-old men) recovered from COVID-19 were reported [32]. Finally, also the oldest woman in the world, Sister Andre, born Lucille Randon, tested positive for the coronavirus at her retirement home in Toulon, but showed no symptoms and celebrated her 117th birthday [33]. All these data concern the first pandemic wave of 2020 and the first months of 2021. What do these centenarians have in common? That of being born before 1919, and therefore, being more than 101 years old in 2020. This introduces the third question.

Poulain et al. [7] examined deaths during the 2020 COVID-19 pandemic by month and year of birth of Belgian centenarians and whose birth occurred in the years around the outbreak of the H1N1 “Spanish flu” pandemic. They noticed that “older” centenarians died significantly lesser than “younger” centenarians. Moreover, this difference reached a maximum on August 1, 1918, as the discriminating cut-off date of birth. This date coincides with the first reports of victims of the Spanish flu pandemic in Belgium. The lifelong persistence of cross-reactive immune mechanisms has been suggested by the authors as a mechanism that allowed Spanish flu-exposed centenarians to be resistant to SARS-CoV-2 infection 100 years later.

Considering what was observed by Poulain et al. in Belgium [7], Caruso et al. [8] analysed the 2020 mortality data of centenarians previously discussed, dividing centenarians by year of birth (1918) into younger and older. Thus, they were divided into two groups, those who died at ages 100 and 101 (younger) and those who died at ages >101 years (older). The older centenarians of both sexes did not show an increase in mortality related to the pandemic. The older centenarians who died in 2020 were 163, 14 fewer than in 2019. On the other hand, in 2020, 313 subjects of 100 and 101 years died while in 2019 were dead 194 subjects with an increase of 61%, without sex differences. Thus, Sicilian data confirm the results of the Belgian paper on the resilience to COVID-19 of older centenarians, suggesting the likelihood of a link between resistance to SARS-Cov-2 in 2020 and exposure to the 1918 H1N1 pandemic influenza.

## 5. Discussions

The data analysed in the present review concerns mostly 2020 because after this date the epidemiology of COVID-19 was changed by vaccines, making interpretation of the data more difficult. It has been demonstrated that older people, including centenarians, after the third dose responded to the mRNA vaccine with the same efficiency as younger subjects. This happens because the mRNA, in addition to encoding the antigenic epitopes, also acts as an adjuvant, triggering a coordinated immune response [34].

Before the induction of an immune response to SARS-CoV-2, the sequence of events in SARS-CoV-2 infection includes I) the initial steps of virus entry using ACE2 as an entry receptor, with virus entry enhanced by TMPRSS2, which primes the spike protein [35], and II) innate sensing of virus RNA by Toll-like receptor 7 (TLR7) with the production of type 1 interferons (IFN) [36]. Theoretically, sex differences could operate at multiple points along these pathways. ACE2 is an X chromosome-encoded gene that is downregulated by oestrogen, while TMPRSS2 is regulated by androgen receptor signalling [37]. Moreover, TLR7 is more expressed in female immune cells [9] with an increased production of type-1 IFNs by TLR7 ligands [38]. However, to the best of our knowledge, it has not yet demonstrated the role of ACE2, TMPRSS2, and TLR7 in the different outcomes of COVID-19 between men and women. Moreover, an ACE2 variant has been described as able to reduce the risk but not the severity of infection. However, sex was not considered in that study [39].

As discussed by Ligotti et al. [40], the synergistic effects of immunosenescence and inflamm-ageing have a big impact on older people’s immune response to SARS-CoV-2 infection and must be considered when looking for factors influencing mortality rates in COVID-19. Changes in innate immune responses and the inability to mount an effective adaptive immune response, together with a higher pro-inflammatory state, may explain both the lack of control of viral replication and the possible clinical consequences triggered by cytokine storm, also reducing the chances of proper recovery after infection resolution. In this last regard, studies on centenarians who have overcome COVID-19 would be of some interest [41].

Genetic differences between men and women are well known, starting from the X chromosome which contains a high density of immune-related genes and regulatory elements extensively involved in both the arms of immune responses. X-linked genes that either escape inactivation or, instead, are preferentially inactivated might affect the dosage of their expression between the sexes, hence further influencing the sex bias in infectious diseases [42]. Several variants of *IFN α and β receptor subunit 2* gene have been associated with COVID-19 severity [43], A recent meta-analysis has demonstrated that a common genetic risk locus on chromosome 3 is associated with increased risks of both morbidity and mortality [44]. Interestingly, there are several chemokine receptors among the genes at this locus, whose involvement in SARS-CoV-2 infection is likely. This could explain the impressive genetic risk effect linked to this locus tagged by *rs10490770* [44] None of these studies were conducted in COVID-19 patients by sex [45].

However, what is known about the difference in immune-inflammatory responses to infections between men and women, differences contributing to health and lifespan disparities between the two sexes, fits very well with COVID-19 pathophysiological mechanisms. In general, oestrogens have immunoenhancing and anti-inflammatory effects, whereas androgens and progesterone have an immunosuppressive effect. A slower rate of decline of several immunological parameters has been shown to occur in women than in men. With age, 15 times more activation of monocyte-specific loci has been reported in men compared to women, while age-specific B cell loci/genes were significantly inactivated in men and moderately activated in women. Thus, older women have greater genomic activity for acquired immune response cells whereas older men have higher genomic activity for monocytes (but the CD163 scavenger receptor is more expressed on female monocytes) and inflammation [45,46,47,48,49,50,51,52].

On the other hand, it has been shown that the difference in the excess mortality between sexes was not unique to COVID-19, but it was like other respiratory infections. The authors included data from 27 European countries, covering the seasons 2016/17 to 2019/20. In periods with increased excess mortality, such as that represented by the winter circulation of respiratory pathogens, excess mortality increased more for men than for women. This increase occurred with similar magnitudes in influenza epidemics and the SARS-CoV-2 pandemic. Thus, the observed sex differences in COVID-19-associated deaths are not a specific feature of the COVID-19 pandemic but are associated with the excess mortality typical of infectious diseases [53]. However, it should be remembered that the data from the Dutch study discussed above led the authors to suggest the presence of additional COVID-19 mechanisms responsible for a higher ratio of male-to-female mortality [13].

In a previous paragraph, we have briefly discussed the pathophysiological differences between men and women that may influence COVID-19 outcome i.e., the role of biological sex. Differences in response to COVID-19 between men and women may be due not only to biological differences (i.e., sex differences) but also to the so-called “socially constructed sex” (i.e., gender differences) [54]. It should be emphasized that there is also a role for gender which determines different behaviours between men and women which might influence exposure to the virus with different impacts on the infection [2,19,45,46,48,50,52,54].

Gender interacts with biological factors in the development and outcomes of the immune response. Differences both in the family and at work in perceived roles of the different sexes will be responsible for different patterns of exposure to antigens. Furthermore, there may also be differences in micronutrient intake and access to preventive healthcare facilities. Health promotion knowledge correlates with educational attainment, which is represented differently between males and females in many cultures, and in the Western world across generations. In countries without a universal health system, access to health care will depend on families’ prioritization strategies to cope with economic constraints, resulting in a limitation to some specific treatments with possible side effects on the immune response. Reproductive issues condition both exposure and outcomes of immune responses, acting both as potential protectors and as potential risk factors for infection [2,19,45,46,48,50,52,54].

Figure 1 summarizes the sex and gender differences above discussed.

Both in Belgian and in Sicilian studies centenarians analysed all together show an excess of mortality with respect to the previous year(s) and older people [7,8]. Centenarians are considered an example of positive biology because they have avoided neonatal mortality, death from infection in the pre-antibiotic era, and age-related disease fatalities, thus living more than 99 years [4,5,6]. This does not mean, however, that they are not frail individuals.

A recent Chinese study [20] performed on 13,859 participants aged 85.8 ± 11.1 years, including 2056 centenarians, shows the prevalence of pre-frailty, frailty, and non-frailty, age-, and sex-stratified: only 5% of centenarians are not frail. Both pre-frailty and frailty were closely correlated to increased risk of mortality, after adjusting for confounders. The association between mortality risk and pre-frailty peaked among centenarians. Thus, centenarians probably died because they were frail.

Concerning the different resilience to COVID-19 of “younger” and “older” centenarians, Poulain et al. [7] speculate that older centenarians who lived during the influenza pandemic may have developed memory lymphocytes capable of recognizing SARS-CoV-2 antigenically related H1N1 epitopes that are still present 100 years later. Indeed, one study performed 90 years after the H1N1 pandemic peak has shown that survivors of the 1918 influenza pandemic possess B lymphocytes (or their progeny) that respond to viral infections, as well as highly functional and neutralizing antibodies, are present throughout the life of the host, even nine or more decades after priming [55]. To the best of our knowledge, however, a cross-reaction between coronaviruses, such as SARS-Cov-2, and influenza myxoviruses has never been demonstrated. Nevertheless, it seems clear that, for centenarians, being born after 1918 implies a different outcome of SARS-CoV-2 infection.

Regarding the possible role of Spanish flu in the resistance to COVID-19, the influenza pandemic of 1918–1919 (influenza A, subtype H1N1, that we now know was a strain of the avian flu) was very virulent, causing symptomatic infections in one-third of the inhabitants of the United States and killing about 0.6% of the total population (550,000 people). In the world, the Pandemic killed more than 20 million people, more than in World War I. The flu itself was a mild “3-day fever” with full recovery and low mortality. The lethality was due to secondary bacterial infections causing severe pneumonia, particularly among pregnant women [56]. It is relevant to our discussion that there are indications of long-term impairments in individuals who were prenatally exposed to the virus.

Cohorts born during pandemics have suffered socioeconomic and physical consequences throughout their lives. Indeed, data from the 1960–1980 United States Decennial Census indicate that cohorts that were in utero during the pandemic showed a reduced level of education, increased rates of physical disability, as well as lower income and lower socioeconomic status than other birth cohorts [57].

According to National Health Interview Surveys of 1982–1996, then, the 1919 birth cohort, most of which was exposed in the uterus to Spanish flu and co-infections, had accelerated cardiovascular ageing with an excess of 5% cardiovascular disease at ages 63–77 years. Furthermore, at the time of conscription for World War II, men born in 1919 were approximately 0.127 cm shorter than the surrounding cohorts. This effect, while small, is highly significant and falls in line with the period of rising heart disease. All these alterations could be linked to the production of glucocorticoids and interleukin-6 by the mother in response to the stress of H1N1 infection with possible epigenetic consequences [58].

Finally, a study conducted in Japan also demonstrated the long-term effects of HIN1 infection; exposure to influenza during foetal life has had significant adverse effects on height development in children from elite backgrounds. The authors suggested that exposure may also have changed, in adulthood, their socioeconomic outcomes, as well as their health. Exposure to the influenza virus during foetal life reduced the height of girls and boys by approximately 0.14 cm and 0.28 cm, respectively [59].

All of these are epigenetic effects of the virus. One possible hypothesis is that these epigenetic effects also affect the immune system.

Exposure to some microbial components can train myelomonocytic cells to develop enhanced effector function against microbial agents. This can occur at the level of both hematopoietic stem cells and mature macrophages. Epigenetic and metabolic reprogramming form the cellular basis of trained immunity. Trained myeloid cells show increased production of cytokines, chemokines, and fluid-phase pattern recognition molecules as well as a greater killing ability. Moreover, these cells become better suited for triggering acquired immune responses [60]. Accordingly, a recent study shows a negative association between the quadrivalent inactivated influenza vaccine and COVID-19 incidence. So, flu vaccination may induce a trained immunity program able to reduce systemic inflammation and regulate transcriptional activity and cytokine production of blood immune cells [61]. Thus, the 1918 H1N1 influenza virus immunity might be thought to confer some protection against SARS-CoV-2 infection. However, it must be underlined that people imprinted with the H1N1 antigen/epitope set due to H1N1 influenza virus infection in childhood were protected later in life only from infections with related viruses such as H5N1 but not from infections with the more distantly related viruses as H3N2 [62].

Our hypothesis, supported by our ongoing studies on the immunology of longevity [63], is that that “older” centenarians are more selected than “younger” ones, and therefore have a more efficient immune system. It has been suggested that a particular intestinal microbiota may also contribute to this efficiency. Older centenarians would indeed have a microbial ecosystem which, although harbouring opportunistic and allochthonous bacteria, is enriched in Akkermansia and Bifidobacterium, known to be associated with immunomodulation, and protection from inflammation. However, we do not know what the cause-and-effect relationships are [64]. All this does not rule out the possibility that the Spanish flu acted as an event able to select newborns/children with a more robust immune system.

In the three epidemiological studies (Lombard, Belgian and Sicilian) on centenarian deaths discussed above, mortality only related to COVID-19 was not analysed, but total mortality was analysed. Indeed, this is a limitation but also a strength. In fact, an increase in general mortality due to the lack of response to other clinical problems cannot be excluded, due to the strong pressure exerted by the pandemic on Health Services. However, this explanation is not completely convincing, because the mortality of older centenarians did not increase (at least in the Sicilian and Belgian studies). Theoretically, it would have been more specific to analyse only confirmed deaths from COVID-19, but it should be considered that in centenarians only a small fraction of deaths due to infection have been reported as such. On the other hand, some of the COVID-19 mortality may have been wrongly attributed to other causes, in the absence of clear typical manifestations since swabs were only partially available.

## 6. Conclusions

Thus, we fully answered to questions we asked ourselves. The answer was positive for the first question as we demonstrated that women were more resilient than men, although there are conflicting findings regarding centenarian men, whereas the answer was negative for the second question since centenarians did not die less than usual. Finally, we answered positively to the third question as, in the first pandemic wave of 2020, older centenarians (>101 years old, born before 1919) were more resistant to SARS-CoV-2 than younger centenarians (100 and 101 years old, born after 1918). Moreover, in the previous paragraph, we examined the possible mechanisms.

As regards the data on centenarians, beyond the hypotheses discussed, this interesting question remains: is the secret of longevity in the efficiency of the immune system? In understanding the role of the immune system in achieving longevity, it must be considered that the immune system of old people has been studied more extensively and in-depth than all other systems and organs due to the ease of ex vivo studies. Therefore, it cannot be easy to assert that the secret of longevity is hidden in the immune system. On the one hand immunosenescence, on the other hand, the maintenance of a relatively good immune response can only be part of the deterioration or, respectively, of the general good functioning of the organism, regulated by factors other than immune ones (brain, endocrine system regulated by the brain, etc.). However, poor and good immune cell function could play a pivotal role in ageing and longevity, respectively. Therefore, the immune system plays a very important role in longevity, which does not however exclude the participation of other systems or organs.

## Figures and Tables

**Figure 1 ijms-24-02638-f001:**
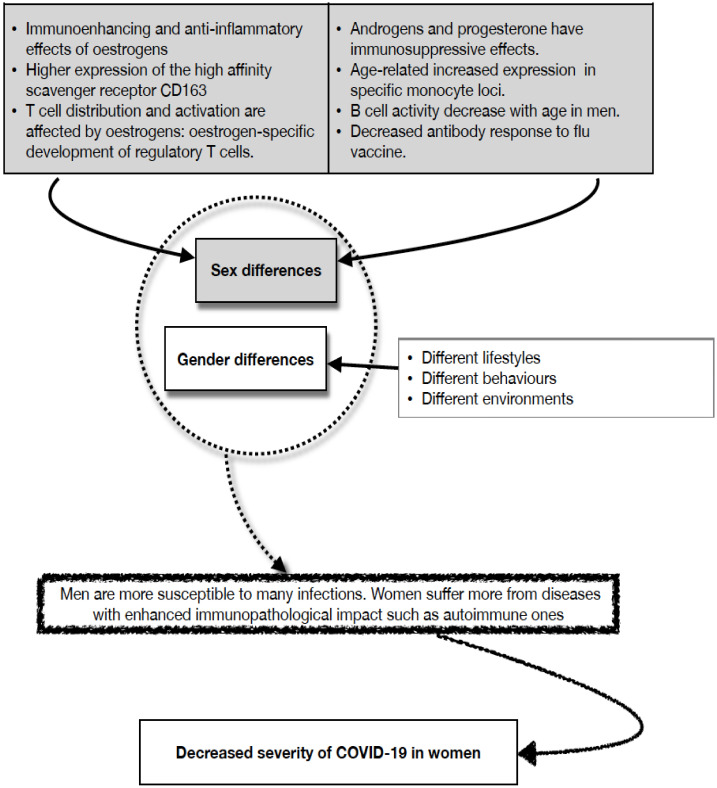
Immune responses differ in men and women The strength and the kind of immune responses are different between males and females (for the references see text).

## Data Availability

Not applicable.

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
