# Peer review of "Role of Sex and Age in Fatal Outcomes of COVID-19: Women and Older Centenarians Are More Resilient"

_ijms, 2023, doi:10.3390/ijms24032638_

Round 1

Reviewer 1 Report

I reviewed manuscript ijms-2152016 by Caruso et al. The authors reviewed the existing literature and they suggested that women and people born before 1918 are more resilient to COVID-19 and present more favorable outcomes compared to men and younger centenarians respectively. The most important disadvantages of the manuscript are the following:

1) The authors included only studies conducted till 2020. They explained that they did not include later studies due to the fact that the vaccine availability and the overall percentage of vaccinated adults would affect the epidemiology and would undermine the interpretation of the data. However, in the later stages of the pandemic it became apparent that the differences between genders regarding mortality were not so large. After all, certain reports indicated that the difference in the excess mortality between genders was not unique for COVID-19 and it is similar to other respiratory infections (Nielsen, J., Nørgaard, S.K., Lanzieri, G. et al. Sex-differences in COVID-19 associated excess mortality is not exceptional for the COVID-19 pandemic. Sci Rep 11, 20815 (2021). https://doi.org/10.1038/s41598-021-00213-w). The authors should make a comment about this in the discussion section and include the above reference.

2) Although the hypothesis that people who were born before 1918 might bear cross-reactive antibodies and therefore be more resilient to COVID-19, than "younger centenarians", the data concerning this population should be interpreted with caution, since many aged people were treated at home and were not included in the studies presented in the current manuscript. This should be mentioned in the "discussion" section.

3) The possible association of viral entry and inflammation with certain phylogenetic characteristics explained in the second paragraph of the "discussion" section (lines 245-251) needs to be supported by more references.

4) The text boxes in figure 1 should be summarized in smaller sentences.

5) Few grammar-vocabulary errors are present throughout the text. 

Author Response

Referee I

I reviewed manuscript ijms-2152016 by Caruso et al. The authors reviewed the existing literature and they suggested that women and people born before 1918 are more resilient to COVID-19 and present more favorable outcomes compared to men and younger centenarians respectively. The most important disadvantages of the manuscript are the following:

Thanks for your suggestions that have improved our paper

1) The authors included only studies conducted till 2020. They explained that they did not include later studies due to the fact that the vaccine availability and the overall percentage of vaccinated adults would affect the epidemiology and would undermine the interpretation of the data. However, in the later stages of the pandemic it became apparent that the differences between genders regarding mortality were not so large. After all, certain reports indicated that the difference in the excess mortality between genders was not unique for COVID-19 and it is similar to other respiratory infections (Nielsen, J., Nørgaard, S.K., Lanzieri, G. et al. Sex-differences in COVID-19 associated excess mortality is not exceptional for the COVID-19 pandemic. Sci Rep 11, 20815 (2021). https://doi.org/10.1038/s41598-021-00213-w). The authors should make a comment about this in the discussion section and include the above reference.

We have added the reference in Discussion and commented about that as it follows

On the other hand, it has been shown that the difference in the excess mortality between sexes was not unique to COVID-19, but it was like other respiratory infections. The authors included data from 27 European countries, covering the seasons 2016/17 to 2019/20. In periods with increased excess mortality, such as that represented by the winter circulation of respiratory pathogens, excess mortality increased more for men than for women. This increase occurred with similar magnitudes in influenza epidemics and the SARS-CoV-2 pandemic. Thus, the observed sex differences in COVID-19-associated deaths are not a specific feature of the COVID-19 pandemic but are associated with the excess mortality typical of infectious diseases [53]. However, it should be remembered that the data from the Dutch study discussed above led the authors to suggest the presence of additional COVID-19 mechanisms responsible for a higher ratio of male to female mortality [13].

2) Although the hypothesis that people who were born before 1918 might bear cross-reactive antibodies and therefore be more resilient to COVID-19, than "younger centenarians", the data concerning this population should be interpreted with caution, since many aged people were treated at home and were not included in the studies presented in the current manuscript. This should be mentioned in the "discussion" section.

As mortality from COVID-19 as excess mortality has been studied in the three epidemiological papers, we have added the specific limitations of our study at the end of the discussion as it follows

In the three epidemiological studies (Lombard, Belgian and Sicilian) on centenarian deaths discussed above, mortality only related to COVID-19 was not analysed, but total mortality was analysed. Indeed, this is a limitation but also a strength. In fact, an increase in general mortality due to the lack of response to other clinical problems cannot be excluded, due to the strong pressure exerted by the pandemic on the Health Services. However, this explanation is not completely convincing, because the mortality of older centenarians did not increase (at least in the Sicilian and Belgian studies). Theoretically, it would have been more specific to analyze only confirmed deaths from COVID-19, but it should be considered that in centenarians only a small fraction of deaths due to infection have been reported as such. On the other hand, some of the COVID-19 mortality may have been wrongly attributed to other causes, in the absence of clear typical manifestations since swabs were only partially available.

3) The possible association of viral entry and inflammation with certain phylogenetic characteristics explained in the second paragraph of the "discussion" section (lines 245-251) needs to be supported by more references.

Done, as it follows

Before the induction of immune response to SARS-CoV-2, the sequence of events in SARS-CoV-2 infection includes I) The initial steps of virus entry using ACE2 as an entry receptor, with virus entry enhanced by TMPRSS2, which primes the spike protein [35] II) Innate sensing of virus RNA by Toll-like receptor 7 (TLR7) with the production of type 1 interferons (IFN) [36]. Theoretically, sex differences could operate at multiple points along these pathways. ACE2 is an X chromosome-encoded gene that is downregulated by oestrogen, while TMPRSS2 is regulated by androgen receptor signalling [37]. Moreover, TLR7 is more expressed in female immune cells [9]  with an increased production of type-1 IFNs by TLR7 ligands [38].However, to best of our knowledge it has not yet demonstrated a role of ACE2, TMPRSS2 and TLR7 in the different outcomes of COVID-19 between men and women. Moreover, an ACE2 variant has been described able to reduce the risk but not the severity of infection. However, sex was not considered in that study [39].

4) The text boxes in figure 1 should be summarized in smaller sentences.

Done.

5) Few grammar-vocabulary errors are present throughout the text.

Done

PLEASE NOTE THAT WE HAVE CHANGED SLIGHTLY THE TITLE FOLLOWING THE COMMENT OF REFEREE II ON THE FIRST QUESTION REPORTED IN THE ABSTRACT (Also, some sentences of the text have been changed to avoid plagiarism)

This question differs from the title of the article, which asks whether women are more resilient. They would have to be concordant.

Response

You are right but we should also rewrite several sentences in the text. So, we preferred to change the title, making it more incisive.

Reviewer 2 Report

This paper is an interesting review that brings together different studies related to the influence of age (especially in centenarians) and sex on coronavirus mortality.
Although the aim of the study is laudable, to answer the three questions or objectives posed, the methodology followed is questionable.
As this is a review of the literature or narrative review, it should include a methodology section describing the research questions, the search terms used, the databases consulted, the criteria used to select the articles, etc.
In the absence of this Methodology section, it is not clear, for example, why 14 articles (self-citations) of the authors have been included.
The presentation of the results is not clear either. There is a lot of redundant information that does not provide relevant information to answer the research objectives.
The conclusions do not answer the objectives of the study.
I have added other contributions in the form of comments in the attached file.

Kind regards

Author Response

Referee Ii

This paper is an interesting review that brings together different studies related to the influence of age (especially in centenarians) and sex on coronavirus mortality. Although the aim of the study is laudable, to answer the three questions or objectives posed, the methodology followed is questionable.

As this is a review of the literature or narrative review, it should include a methodology section describing the research questions, the search terms used, the databases consulted, the criteria used to select the articles, etc.

In the absence of this Methodology section, it is not clear, for example, why 14 articles (self-citations) of the authors have been included.

Thanks for your suggestion. We have added the following paragraph (obviously the methodology must and/or can only explain the choice of papers related to the effect of age and gender on COVID-19 mortality, not the choice of papers related to immunosenescence and longevity biology)

To carry out this narrative review, we queried the PUBMED database. Regarding mortality due to sex differences, we included the terms Sex Differences, COVID-19, and Death (or Mortality), limiting ourselves to meta-analysis and review to avoid analysing a plethora of studies, obtaining seven entries. Of these, only three were relevant to our research. Analysing, then, both the papers cited by these three articles and the list of papers citing them, we found three more pertinent reviews (plus two other papers of some interest). Due to the overlap of the analysed data, only three reviews are discussed in the following paragraphs [9-11]. Moreover, since Italy was the first country affected by SARS-CoV-2 in the western world, we have added a paper on Italian data in the first months of the epidemic [12]. We have also added a paper on a study carried out in the Netherlands because it is very informative as it shows comparisons with other infectious diseases [13]. As regards the mortality of centenarians, we entered the terms Centenarian, COVID-19, and Death (or Mortality), obtaining 8 entries. Of these, only 4 were pertinent to our research. Analysing, then, both the papers cited by these four articles and the list of papers citing them, we found a fifth paper pertinent. All five articles [7,8, 14-16]-are discussed in the following paragraphs.

The presentation of the results is not clear either. There is a lot of redundant information that does not provide relevant information to answer the research

objectives.

Thanks. Following your suggestions reported in the text you commented on (that we upload with  our response to your comments), we have improved our paper.

The conclusions do not answer the objectives of the study.

According to your suggestions, we have changed the first paragraph and the first line of the second paragraph of Conclusion, as it follows

I have added other contributions in the form of comments in the attached file.

Thanks, As previously stated we upload the file with our replies.

Please note some sen

Round 2

Reviewer 2 Report

The authors have carried out a very thorough review, incorporating the contributions made in an appropriate manner.
After the review, there are small areas for improvement:

-        In the questions to be answered (in the summary and in the text) it would be more appropriate to ask: "Have women been more resilient than men?".

-        The text should clearly describe what is meant by younger centenarians, older centenarians and ultracentenarians (Which is the difference between older centenarians and ultracentenarians?). It should also be clarified in the abstract of the article.

-        Conclusions: The paragraph you have introduced to answer the objectives of the study is necessary, but not well understood. It would be more understandable if the paragraph were written in the affirmative, without repeating the questions posed.

-        Other more specific considerations:

·        Line 160:  This characteristic has been observed in different studies worldwide and it is also evident in Northern Italy, including Lombardy, and in North-East of Italy, where the women in “Centenarians in Trieste Study” suffer greater morbidity, but they live longer than men reaching 163 90% prevalence [14, 18, 19].

o    The reference should be added here in order to differentiate with the study carried out in North-East of Italy

·        Line 195: These data confirm that women over ninety are more resilient to the virus than men as suggested by anecdotal findings discussed below [22,23] , but not by the study conducted in Lombardy.

o    Add reference

·        Line 304: Then, in the previous discussed Sicilian study [8]

o    This phrase is not connected to other parts of the text.

Best regards

Author Response

The authors have carried out a very thorough review, incorporating the contributions made in an appropriate manner. 
After the review, there are small areas for improvement:

In the questions to be answered (in the summary and in the text) it would be more appropriate to ask: "Have women been more resilient than men?".

DONE

 Changed 3 times: in the abstract, in the headline and at the end of the second chapter

The text should clearly describe what is meant by younger centenarians, older centenarians and ultracentenarians (Which is the difference between older centenarians and ultracentenarians?). It should also be clarified in the abstract of the article.

DONE.

You are right. Now I think we have clarified the meaning in the abstract and text. Moreover, to avoid misunderstandings we have eliminated the term ultracentenarians. In the meantime, we have harmonized both the abstract and the text (and the related headline of chapter 4) both in the questions and in the answers, always comparing older to younger; we have also specified that the age of the centenarians refers to the 2020 pandemic wave

Conclusions: The paragraph you have introduced to answer the objectives of the study is necessary, but not well understood. It would be more understandable if the paragraph were written in the affirmative, without repeating the questions posed.

DONE

Thus, we fully answered to questions we asked ourselves. The answer was positive for the first question as we have demonstrated that women were more resilient than men, although there are conflicting findings regarding centenarian men, whereas the answer was negative for the second question since centenarians did not die less than usual.  Finally, we answered positively to the third question as, in the first pandemic wave of 2020, older centenarians (>101 years old, born before 1919) were more resistant to SARS-CoV-2 than younger centenarians (100 and 101 years old, born after 1918).

Other more specific considerations:

Line 160:  This characteristic has been observed in different studies worldwide and it is also evident in Northern Italy, including Lombardy, and in North-East of Italy, where the women in “Centenarians in Trieste Study” suffer greater morbidity, but they live longer than men reaching 163 90% prevalence [14, 18, 19].

 The reference should be added here in order to differentiate with the study carried out in North-East of Italy.

DONE

[18]·      

  Line 195: These data confirm that women over ninety are more resilient to the virus than men as suggested by anecdotal findings discussed below [22,23] , but not by the study conducted in Lombardy.

Add reference

DONE

[14]

Line 304: Then, in the previous discussed Sicilian study [8]

This phrase is not connected to other parts of the text.

DONE

Considering what observed by Poulain et al. in Belgium [7], Caruso et al. [8] analysed the 2020 mortality data of centenarians previously discussed, dividing centenarians by year of birth (1918) into younger and older. Thus, they were divided in two groups, those who died at ages 100 and 101 (younger) and those who died at ages >101 years (older). The older centenarians of both sexes did not show an increase in mortality related to the pandemic.